# Δ9-Tetrahydrocannabinol Prevents Mortality from Acute Respiratory Distress Syndrome through the Induction of Apoptosis in Immune Cells, Leading to Cytokine Storm Suppression

**DOI:** 10.3390/ijms21176244

**Published:** 2020-08-28

**Authors:** Amira Mohammed, Hasan F.K. Alghetaa, Kathryn Miranda, Kiesha Wilson, Narendra P. Singh, Guoshuai Cai, Nagireddy Putluri, Prakash Nagarkatti, Mitzi Nagarkatti

**Affiliations:** 1Department of Pathology, Microbiology and Immunology, School of Medicine, University of South Carolina, Columbia, SC 29208, USA; amira.mohammed@uscmed.sc.edu (A.M.); hasan.alghetaa@uscmed.sc.edu (H.F.K.A.); kathryn.miranda@uscmed.sc.edu (K.M.); Kiesha.wilson@uscmed.sc.edu (K.W.); narendra.singh@uscmed.sc.edu (N.P.S.); prakash@mailbox.sc.edu (P.N.); 2Department of Environmental Health Sciences, Arnold School of Public Health, University of South Carolina, Columbia, SC 29208, USA; gcai@mailbox.sc.edu; 3Dan L. Duncan Cancer Center, Advanced Technology Core, Alkek Center for Molecular Discovery, Baylor College of Medicine, Houston, TX 77030, USA; putluri@bcm.edu; 4Department of Molecular and Cell Biology, Baylor College of Medicine, Houston, TX 77030, USA

**Keywords:** acute respiratory distress syndrome, Δ9-tetrahydrocannabinol, staphylococcal enterotoxin B, cytokine storm, apoptosis

## Abstract

Acute Respiratory Distress Syndrome (ARDS) causes up to 40% mortality in humans and is difficult to treat. ARDS is also one of the major triggers of mortality associated with coronavirus-induced disease (COVID-19). We used a mouse model of ARDS induced by Staphylococcal enterotoxin B (SEB), which triggers 100% mortality, to investigate the mechanisms through which Δ9-tetrahydrocannabinol (THC) attenuates ARDS. SEB was used to trigger ARDS in C3H mice. These mice were treated with THC and analyzed for survival, ARDS, cytokine storm, and metabolome. Additionally, cells isolated from the lungs were used to perform single-cell RNA sequencing and transcriptome analysis. A database analysis of human COVID-19 patients was also performed to compare the signaling pathways with SEB-mediated ARDS. The treatment of SEB-mediated ARDS mice with THC led to a 100% survival, decreased lung inflammation, and the suppression of cytokine storm. This was associated with immune cell apoptosis involving the mitochondrial pathway, as suggested by single-cell RNA sequencing. A transcriptomic analysis of immune cells from the lungs revealed an increase in mitochondrial respiratory chain enzymes following THC treatment. In addition, metabolomic analysis revealed elevated serum concentrations of amino acids, lysine, n-acetyl methionine, carnitine, and propionyl L-carnitine in THC-treated mice. THC caused the downregulation of miR-185, which correlated with an increase in the pro-apoptotic gene targets. Interestingly, the gene expression datasets from the bronchoalveolar lavage fluid (BALF) of human COVID-19 patients showed some similarities between cytokine and apoptotic genes with SEB-induced ARDS. Collectively, this study suggests that the activation of cannabinoid receptors may serve as a therapeutic modality to treat ARDS associated with COVID-19.

## 1. Introduction

Staphylococcal enterotoxin B (SEB) is a superantigen that promotes massive inflammation by triggering a large proportion of T cells expressing certain Vβ T cell receptors [1]. Depending on the route of exposure, SEB can promote toxic responses, leading to food poisoning, toxic shock syndrome, or acute lung injury (ALI). The inhalation of SEB promotes ALI, a life-threatening condition that is characterized by leukocyte infiltration, pro-inflammatory cytokine production, and the breakdown of the lung barrier. In a C3H mouse model, we have previously shown that dual-dose exposure to SEB involving the intranasal route followed by systemic exposure triggers Acute Respiratory Distress Syndrome (ARDS), leading to 100% mortality [2,3]. In humans, ARDS can be triggered by infectious agents that trigger a life-threatening condition characterized by severe pulmonary inflammation, poor oxygenation, and respiratory failure [4]. Because there are no specific and effective treatment modalities, up to 40% of ARDS patients die. Interestingly, patients with a severe form of novel coronavirus disease 2019 (COVID-19) were found to exhibit ARDS, cytokine storm, and pulmonary failure [5,6].

Δ9-tetrahydrocannabinol (THC) is a psychoactive cannabinoid derived from *Cannabis sativa* that has potential therapeutic value for pain relief, control of nausea and vomiting, appetite stimulation, and its anti-inflammatory properties [7]. Indeed, a previous report from our laboratory showed that exposure to THC prior to the administration of SEB can prevent SEB-induced ARDS and associated mortality through the miRNA (miR) regulation of regulatory T cells [8]. In the current study, we tested whether THC administration after exposure to SEB would prevent SEB-mediated ARDS; to further understand the mechanisms, we used single-cell RNA sequencing (scRNA-seq) of cells isolated from the lungs.

Apoptosis is a form of highly regulated programmed cell death that can be triggered through the intrinsic (mitochondrial) or extrinsic (death receptor-mediated) signaling pathways [9]. Here, we attempted to clarify the role of THC in inducing apoptosis in immune cells as a mechanism of attenuation of ARDS. Our laboratory and others have indicated that THC can induce apoptosis in different cell types [10,11,12]. In T cell leukemia (Jurkat cell line) cells, for instance, a study from our lab showed that THC can induce apoptosis via cross talk between intrinsic and extrinsic pathways [13]. Thus, in the current study based on single-cell RNA sequencing data, we determined whether THC induces apoptosis in activated immune cells in the lungs following SEB-induced ARDS and, if so, whether it was through the death receptor or mitochondrial pathway.

Single-cell RNA sequencing (scRNA-seq) is a relatively novel and powerful technique for quantitating the transcriptome of various cell types in tissues based on molecular characteristics rather than on the morphology or proteins in cells [14]. In the current study, we used this technology to unveil the precise cells and molecular signatures that may be involved in the THC-mediated induction of apoptosis. Furthermore, the data were integrated with our findings from metabolomic analysis of serum metabolites, as well as cellular respiration, mitochondrial function, and metabolism.

In recent years, miRNAs (miRs) have been identified to suppress multiple genes, and thus regulate biological processes [15]. Studies from our lab have shown the ability of THC to suppress neuroinflammation by the downregulation of miR-21, which upregulates *Bcl2L11* [16]. Furthermore, we demonstrated that THC treatment caused the elevation of anti-inflammatory myeloid-derived suppressor cells (MDSCs) through the miR-690-targeted *C*/*EBPα* gene [17] and via miR-34a-targeted *Nos1* [3].

In the current study, we also investigated the role of miRNA in the regulation of apoptosis in immune cells induced by THC to attenuate SEB-mediated ARDS. Our data demonstrated that THC decreased the expression of miR-185-3p in SEB-activated immune cells, thereby promoting the induction of a number of genes related to the mitochondrial pathway of apoptosis, causing an alteration in metabolism of immune cells, leading to the attenuation of inflammation and ARDS.

## 2. Results

### 2.1. THC Treatment Prevents SEB-Induced Pulmonary Damage via a Reduction in Infiltrating Immune Cells

It is well known that dual-dose SEB exposure in C3H mice is fatal due to the massive production of inflammatory cytokines and chemokines that lead to the exponential proliferation of effector T cells and other immune cell phenotypes [18]. Consistent with our earlier published studies, we found that while SEB exposure in mice caused a 100% mortality, treatment with THC led to the 100% survival of the mice [3,19]. Additionally, we found that dual SEB exposure versus naïve mice resulted in a massive infiltration of immune cells in the lungs (Figure 1A). Interestingly, SEB+THC mice showed a noticeable reduction in the abundance of infiltrating cells in the lung parenchyma when compared to SEB+Veh mice (Figure 1A). Electronic cell-substrate impedance sensing (ECIS) was performed to measure the epithelial cell resistance. Our data showed that epithelial cells treated with SEB+THC had a higher resistance than SEB+Veh (Figure 1B). Consequently, SEB+Veh mice, in contrast to naïve mice, had elevated levels of proinflammatory cytokines IFN-γ, IL-1β, IL-2, TNF-α, and IL6 in BALF, while THC treatment led to a significant reduction in these cytokines (Figure 1C). Similar trends for the chemokines, CCL2, CCL5, and CXCL1 were also observed (Figure 1D).

### 2.2. THC Induces Apoptosis through the Mitochondrial Pathway

To determine whether the decrease in lung infiltration was due to induction of apoptosis, we performed Tunel staining by flow cytometry. Our data showed that THC induced apoptosis in lung-infiltrating mono-nuclear cells, MNCs (Figure 2A). THC also led to a reduction in the mitochondrial membrane potential by DiOC(6)3 staining (Figure 2B). To confirm our in vivo results, we examined apoptosis in vitro. To this end, we activated naïve splenocytes with SEB (1 μg/mL) for 72 h in the presence of either 10 uM THC or Veh. The cells were then collected and stained with anti-CD3 and DiOC(6)3 and, likewise, THC led to increased apoptosis and the loss of the mitochondrial membrane potential in SEB-activated T cells (Figure 2C). Furthermore, we tested the direct effect of THC on the proliferative capacity of immune cells in vitro by performing a 3H-thymidine incorporation assay, specifically by stimulating naïve splenocytes with SEB and then treating them with THC or Veh for 72 h. As expected, SEB activation led to increased proliferation when compared to non-activated T cells, while 10 μM of THC reduced the SEB-induced proliferation (Figure 2D). We further analyzed lung-infiltrating MNCs using mouse transcriptome arrays, and found that THC increased the expression of genes related to apoptosis, such as caspases (*Casp1*, *Casp14*); the release of mitochondrial cytochrome c, including mitochondrial apoptogenic protein 1 or cytochrome c oxidase assembly factor 8 (*Apopt1* or *Coa8*) and *Coa4*; other mitochondrial cytochrome c oxidases (*Cox6a1*, *Cox7a1*, *Cox18*); apoptosis-inducing factor 2-homologous mitochondrion-associated inducer of death (*Aifm2*); as well as autophagic cell death comprising autophagy-related 5 and 10 (*Atg5* and *Atg10*) (Figure 2E). These studies suggested THC mediated the induction of apoptosis and autophagic cell death by altering the cytochrome c oxidases of the mitochondrial electron transport chain in MNCs infiltrating the lung in SEB-induced ARDS.

### 2.3. Pulmonary scRNA-seq Shows a Reduction in Inflammatory Components in SEB+THC-Treated Mice

In order to elucidate the precise cells and the genes that may be altered in the lungs, we performed scRNA-seq. Whole lung tissue was collected from SEB-administered mice treated with either THC or Veh and processed to a single cell suspension for scRNA-seq. t-Distributed Stochastic Neighbor Embedding (t-SNE) plots showed that the distribution of the major subpopulations of immune cells in ARDS were alveolar macrophages, *Mac1a* and *Mac1b* macrophages, neutrophils, CD4+ and CD8+ T lymphocytes, and natural killer (NK) and NKT cells (Figure 3A). The enrichment of these inflammatory populations was higher in Veh-treated mice than in THC-treated mice (Figure 3B). Interestingly, a scRNA-seq analysis showed an increase in several genes in the SEB+THC versus SEB+Veh groups, which included mitochondria-associated apoptosis regulatory genes. This comprised the elevated expression of *Bad* in the CD4+ and CD8+ T cells, NKT cells, as well as *Mac1a* and *Mac1b* macrophages (Figure 3C); *BCL2* Associated X Apoptosis Regulator (*Bax*) in CD8+ T cells and NKT cells (Figure 3D); *Cox4i1c* in CD4+ T cells, neutrophils, NK cells, B cells, and alveolar and *Mac1b* macrophages (Figure 3E); *Apopt1* in CD4+ and CD8+ T cells, NKT cells, and NK cells (Figure 3F); as well as *Casp3* in CD4+ and CD8+ T cells and *Mac1b* macrophages (Figure 3G) in the SEB+THC treated group when compared to SEB+Veh mice. Additionally, an scRNA-seq analysis showed that THC increased the expression of genes involved in the metabolic reprogramming of immune cells, specifically genes encoding the mitochondrial solute carrier family of proteins (*Slc25*) including a phosphate carrier protein, *Slc25a3*, and an amino acid or iron carrier protein, *Slc25a39* (Figure 3H).

### 2.4. THC-Mediated Amelioration of ARDS Is Associated with Altered T Cell Metabolism

Given that THC induced apoptosis via the mitochondrial pathway and affected mitochondrial solute transporter proteins, we next performed a real-time cell metabolic analysis of purified T cells. For this purpose, using Seahorse technology we determined the oxygen consumption rate (OCR), including first the basal mitochondrial and non-mitochondrial respiration, and, following the sequential addition of oligomycin, an ATP synthase inhibitor, the ATP-linked respiration and proton leak. Next, we treated the cells with carbonyl cyanide-4-(trifluoromethoxy)phenylhydrazone (FCCP), which mimics the physiological stimulation of the respiratory chain, causing the oxidation of all substrates including sugars, fats, and amino acids to obtain maximal respiratory capacity. Finally, we added Rotenone/Antimycin A and respiratory chain complex I and III inhibitors, respectively, to determine the reserve capacity. Our studies on OCR showed that T cells from the SEB+THC group had decreased basal respiration, proton leak, ATP-linked respiration, maximal respiratory capacity, and reserve capacity when compared to the SEB+Veh group (Figure 4A). We next estimated the dependency of these activated T cells on glucose versus fatty acids as main substrates for their metabolic functions. In order to determine the glucose dependency, we examined the OCR by blocking glucose oxidation in the presence of UK5099, an inhibitor of mitochondrial pyruvate carrier, followed by the blocking of both fatty acid and glutamine oxidation using Etomoxir, a carnitine palmitoyl transferase 1 inhibitor, and bis-2-(5-phenylacetamido-1,2,4-thiadiazol-2-yl)ethyl sulfide (BPTES), a glutaminase inhibitor. Furthermore, the β-oxidation dependency was studied by examining OCR in the presence of etomoxir, which blocks fatty acid metabolism, followed by the treatment of the cells with BPTES and UK5099, which block both amino acid and glucose oxidation. Interestingly, we found that while the OCR was decreased significantly in SEB+THC cells, the metabolic activity of the THC-treated cells was independent of glucose utilization (Figure 4B). However, these THC-treated T cells trended towards being dependent on the β-oxidation pathway to fulfill their energy demands (Figure 4C). In contrast, the SEB+Veh-treated cells were dependent on glucose oxidation but independent of fatty acid oxidation for their energy demands.

### 2.5. Metabolomic Profiling

Furthermore, the metabolomic profile of serum showed that SEB+THC mice had elevated levels of carnitine pathway metabolites, which are displayed in a heat map (Figure 4D), when compared to the SEB+Veh group. Specifically, the metabolomic analysis showed a significant increase in the serum concentrations of propionyl-L-carnitine (PLC) (Figure 4E), free carnitine (Figure 4F), lysine (Figure 4G), and n-acetyl methionine (Figure 4H) in SEB+THC mice, indicating that there was dysregulation in carnitine metabolism pathway following SEB exposure. Carnitine is made from amino acids, lysine, and methionine. Carnitine transports long-chain fatty acids from the cytosol to the mitochondrial matrix, and is therefore important for β-oxidation. Carnitine can be converted to PLC. Interestingly, PLC has been shown to promote the induction of apoptosis [20]. To test if the effect of THC on metabolism may be contributing to the increased cell death in infiltrating immune cells, we activated naïve splenocytes with SEB (1 μg/mL) for 72 h in the presence of PLC (200 μM). The data showed that PLC-treated cells had an increased loss in mitochondrial membrane potential and increased apoptosis (Figure 4I). Next, a 3H-thymidine incorporation assay confirmed that PLC had no significant effect on the proliferation of naïve splenocytes (Figure 4J). However, the addition of PLC led to a significant suppression of proliferation in SEB-activated splenocytes (Figure 4J).

### 2.6. THC-Mediated Apoptosis of SEB-Activated T Cells Is Epigenetically Regulated

We performed a miR microarray and found a number of dysregulated miRs in these two groups of mice. Ingenuity pathway analysis (IPA) was used to predict the most common pathways involved in modulating the immune cell metabolism and proliferation capacity of lung-infiltrating MNCs. We found that miR-185-3p was downregulated in SEB+THC versus SEB+Veh mice. Per the IPA analysis, miR-185-3p promoted apoptosis induction and inhibited the *NFkB* signaling (Figure 5A). Next, we validated the downregulation of miR-185 expression in lung-infiltrating MNCs in SEB+THC in the SEB+VEH group using qRT-PCR (Figure 5B). To confirm the alignment existence between the miR-185 and 3′UTR region of target genes, microRNA.org was used (Figure 5C). Furthermore, qRT-PCR validated the upregulation of miR-185 target genes that regulate apoptosis, including *Bad* (Figure 5D), *Bax* (Figure 5E), *Hrk* (Figure 5F), *Runx3* (Figure 5G), and *Cox4* (Figure 5H). Additionally, we found that *NKIRAS2*, encoding an inhibitor of *NFĸB* signaling, was elevated in SEB+THC versus SEB+Veh (Figure 5I).

### 2.7. Meta-Analysis of Genes Altered in SEB-Induced ARDS in Mice vs. COVID-19 BALF Human Datasets

Because patients with a severe form of COVID-19 develop sepsis and ARDS, we next correlated our study with two available gene expression datasets from the BALF of human COVID-19 patients. The datasets were examined for cytokine and apoptotic genes similarly dysregulated between SEB+Veh and SEB+THC using scRNA-seq vs. COVID-19 vs. normal control BALF samples. Venn diagram analysis showed that pro-inflammatory cytokines, including *CCL3*, *IL2RG*, and *TNFAIP3*, were upregulated in CD4+ T cells, CD8+ T cells, and NK cells in the SEB-induced ARDS group and in the BALF of patients with the COVID-19 disease (Figure 6A). Additionally, apoptosis pathway genes such as *CoxIV* and *Bax* were down-regulated in the SEB-induced ARDS group and COVID19 patients (Figure 6B).

## 3. Discussion

ARDS is a disorder caused by acute pulmonary inflammation, resulting in severe lung damage. Because currently there are no effective pharmacological therapies to prevent this inflammatory condition, ARDS is characterized by high mortality rates of 36–44% [21]. Risk factors which are associated with the development of ARDS could be genetic predisposition, obesity, chronic alcohol abuse, and chronic liver disease [22,23,24,25]. Many survivors experience a poor quality of life, depression, anxiety, and post-traumatic stress disorder (PTSD) [26,27,28,29]. It is interesting to note that while the majority of COVID-19 patients get milder form of the disease, ~30% get a more severe form of the disease and develop ARDS, characterized by sepsis, cytokine storm, and respiratory and multiorgan failure, and a significant proportion of such patients die [30]. It is likely that severe infection with the novel virus SARS-CoV-2 that causes COVID-19 in the lower respiratory tract may cause dysbiosis, leading to the emergence of pathogenic bacteria such as *Staphylococcus aureus* and *Streptococcus pneumoniae* [31,32] and triggering ARDS. Thus, the current animal studies using SEB-induced ARDS may have some relevance to the ARDS seen in COVID-19.

Interestingly, we found a correlation between the upregulation of inflammatory cytokine genes in both SEB-induced ARDS and COVID-19 disease, which were downregulated with THC treatment in SEB mice [33]. Additionally, our analysis showed that genes in the apoptosis pathway, such as *CoxIV* and *Bax*, were downregulated in both SEB-induced ARDS group and COVID-19 group. These data suggested that THC may be used as an immunosuppressive agent to dampen cytokine storm and promote apoptosis in activated immune cells during COVID-19. These correlations warrant further in-depth study.

SEB is one of the most important toxin threats in bioterrorism and is listed as a biological-warfare agent, which the Centers for Disease Control (CDC) has classified as a category B priority agent [34]. Investigation from our lab found that SEB inhalation caused vascular leak, the massive infiltration of lymphocytes, and cell death in the endothelial cells of the terminal vessels of the lung during ARDS [35].

Cannabis is the marijuana plant that has more than 113 compounds called cannabinoids. THC is generally the most abundant cannabinoid found in Cannabis extracts [36]. THC is also the main psychoactive compound in marijuana, but is known to have anti-inflammatory properties, and can induce apoptosis and autophagy [12,37,38]. In the current study, we found that treatment with THC after exposure to SEB can induce apoptosis in activated immune cells in a SEB-mediated ARDS model.

In a recent study, we found that THC can reduce the inflammation in the lungs by decreasing the infiltrating immune cells, edema, and congestion [39]. Furthermore, THC decreased pro inflammatory cytokines IFN-γ, IL-1β, IL-2, TNF-α, and IL6, which are the most important cytokines that lead to ARDS [3]. We also found a reduction in CCL2, CCL5, and CXCL1 which are chemokines that recruit leukocytes, including T cells such as memory T cells and monocytes [40,41]. In the current study, we observed that THC can induce apoptosis in the MNCs of the lung by the intrinsic pathway. TUNEL staining, which depends on terminal deoxynucleotidyl transferase (TdT)-mediated dUTP nick-end labeling to detect apoptotic DNA fragmentation, was increased significantly after the THC treatment [42,43,44]. DiOC6(3) is a cationic dye which strongly labels mitochondria. The loss of mitochondrial membrane potential (MMP) is among the changes during the early stages of apoptosis, and a decrease in the MMP in apoptotic cells is associated with a reduction in the expression of DiOC6(3), which we observed after THC treatment [45,46]. The dose-dependent decrease in activated T cell proliferation in THC-exposed cells was consistent with its ability to induce apoptosis.

In the current study, we sought to identify dysregulated genes by examining the transcriptome in MNCs from the lungs in SEB+Veh and SEB+THC groups. Interestingly, we found that genes associated with mitochondrial functions, such as Mitochondrial Apoptogenic Protein 1 (*Apopt1*), also known as cytochrome c oxidase assembly factor 8 (*Coa8*); and *Coa4*; mitochondrial cytochrome c oxidases (*Cox6a1*, *Cox7a1*, *Cox18*); and apoptosis-inducing factor 2-homologous mitochondrion-associated inducer of death (*Aifm2*), were upregulated in lung-infiltrating MNCs following THC treatment in SEB-induced ARDS. These genes encode for proteins that localize in the mitochondria, where they stimulate the release of cytochrome c and consequently induce apoptosis [47]. Cytochrome oxidase assembly factor genes encode for proteins that are required for the collection of the terminal cytochrome c complex IV or cytochrome c oxidase of the mitochondrial respiratory chain, which transports electrons to molecular oxygen and contributes to ATP synthesis [48]. In addition, autophagy-related 5 and 10 (*Atg5 and Atg10*) were also upregulated following THC treatment. Inasmuch as autophagy is a process of the removal of damaged organelles and misfolded or aggregated proteins, it plays a critical role in preventing inflammatory diseases [49]. Specifically, it has been demonstrated that naive T cells exiting from the thymus depend on autophagy-related mitochondrial content reduction [50]. Furthermore, the loss of *ATG5* in T cells was shown to result in an inflammatory response and a loss of tolerance [50]. Indeed, the reduction in the expression of *Atg* genes observed in the SEB+Veh group may be a factor in the induction of inflammatory response. These studies suggested the THC-mediated involvement in apoptosis and autophagy, and, importantly, cytochrome c oxidases of the mitochondrial electron transport chain in mononuclear cells infiltrating the lungs following SEB-induced ARDS.

In order to detect genes involved in THC-induced apoptosis in various immune cell subpopulations, we performed scRNA-seq. The mitochondrial pathway of apoptosis is influenced by *Bcl* family members bound to the mitochondrial membrane, including both the anti-apoptotic members, *Bcl-2* and *Bcl-xL* as well as the pro-apoptotic regulatory proteins, Bcl2-associated agonist of cell death (*Bax*) and *Bak*, which can induce apoptosis by forming a pore in the mitochondrial outer membrane for the release of cytochrome c and other pro-apoptotic factors to the cytosol [51]. In contrast, *Bad* heterodimerizes with the anti-apoptotic regulatory protein *Bcl-2* and inactivates it, thereby allowing *Bax*/*Bak* to induce apoptosis. In the cytoplasm, cytochrome c along with adenosine triphosphate (ATP) binds to apoptosis protease activating factor-1 (*APAF-1*) to form a multimeric complex that recruits and activates pro-caspase-9 to caspase 9, which in turn activates caspase 3 and induces apoptosis [52]. Caspase-3 gene was elevated in the CD4+ and CD8+ T cells from the SEB+THC group, which regulates the inhibition of DNA repair and starts DNA degradation [9]. It should be noted that mitochondrial cytochrome c and cytochrome c oxidase subunit IV (COX IV) productivity are early events in the progression of apoptosis onset [53]. Furthermore, *Apopt1*, which was elevated in SEB+THC when compared to SEB+Veh, has also been shown to induce apoptosis independent of *Bax*/*Bak* pore formation [54].

Our studies also found that THC caused an alteration in T cell metabolism. A metabolic analysis of SEB+THC versus SEB+Veh T cells detected a reduction in mitochondrial respiration, specifically in basal respiration, maximal respiratory capacity, proton leak, and spare respiratory capacity, while increasing the dependence on the fatty acid metabolism. This observation indicates that THC stunts cell growth based on a previous study which showed that T cell growth and function requires rapid increases in glycolysis and a decrease in lipid metabolism [55]. During activation, T cells undergo metabolic reprogramming, which is believed to be critical for cells to sustain the biosynthesis of lipids, proteins, and nucleic acids required for cell proliferation. Therefore, increased glycolysis is observed in effector and activated T cells [56].

We also found that THC caused alterations in serum metabolome, specifically elevations in lysine, N-acetyl methionine, carnitine, and propionyl carnitine (PLC). Lysine and methionine are precursors of carnitine, which is found in animal proteins and plays an important role in fat metabolism by transporting long chain fatty acids from cytosol to the mitochondrial matrix, and is therefore useful in the β-oxidation of fatty acids. L-carnitine has been used in the treatment of cardiovascular diseases, end-stage renal diseases, and other diseases [57]. N-acetyl-L-methionine is similar to L-methionine both nutritionally and metabolically [58]. It is used as a dietary supplement. Methionine is an essential amino acid required for normal development in humans. Methionine deficiency leads to a decrease in liver functions. It is also required for cysteine synthesis. While methionine is required for angiogenesis, high levels may lead to an increase in homocysteine, which is an indicator of cardiovascular disease [59]. The presence of excess methionine may also lead to DNA methylation and the induction of cancer [59].

Propionyl-L-Carnitine is a naturally occurring amino acid derived from L-carnitine. PLC is used in the treatment of heart failure and peripheral vascular disease. Importantly, PLC can induce apoptosis through the intrinsic pathway via the activation of *Bax* gene [19]. Our results were consistent with this observation, inasmuch as the addition of PLC triggered apoptosis and decreased cell proliferation.

We have reported that THC treatment following SEB injection alters the expression of miRs in lung-infiltrating immune cells [8] In this study, we focused on miR-185, which was found to be downregulated by THC in lung-infiltrating MNCs of mice exposed to SEB when compared to mice treated with vehicle. Upon a pathway analysis of miRs using IPA, we identified miR-185 in MNCs, which may play a significant role in THC-induced immune suppression via the induction of apoptotic genes *Bad*, *Bax*, Activator of apoptosis harakiri (*Hrk*), and *Apopt1*. miR-185 plays an essential role in in various diseases—for instance, tumor suppression in nasopharyngeal carcinoma, and it also inhibits the invasion, migration, and proliferation of squamous cell carcinoma [60,61]. Furthermore, miR-185 can be therapeutic target for gastric cancer because it can induce apoptosis via the activation of the *Runx3* gene [62]. The inhibition of miR-185 caused *PTEN* induction and *AKT* inhibition, thus promoting apoptosis [63]. miR-185 can also induce apoptosis in lung epithelial cells and a decrease in cell proliferation during hypoxia [64]. Together, our data indicate that THC may have beneficial effects on ARDS by promoting immune cell apoptosis via the downregulation of miR-185-3p.

While our studies suggest that THC may be useful in treating ARDS seen during sepsis, as well as COVID-19, its clinical use may pose some problems because of its psychoactive properties. However, it is worth noting that THC is approved by the FDA to treat nausea and vomiting in cancer patients undergoing chemotherapy and to gain appetite in HIV/AIDS patients. In this context, it is worth noting that cannabidiol (CBD), a non-psychoactive cannabinoid, may be better suited because CBD also exerts anti-inflammatory properties. A recent study showed that CBD can suppress ARDS and cytokine storm induced by Poly(I:C) [65]. While we have also shown that CBD is highly effective against a variety of autoimmune diseases [66,67], it is less potent in suppressing cytokine storm when compared to THC. Clearly additional studies are necessary to compare the efficacy of THC and CBD to treat ARDS and cytokine storm in a variety of clinical models.

## 4. Material and Methods

### 4.1. Mice Grouping and Housing

Adult female C3H/HeJ mice were purchased from the Jackson laboratory (JAX Stock # 000659) (Augusta, ME, USA). All the delivered mice were kept for one week as an acclimatization period prior to performing any experiments. The animals were housed in maximum of 5 mice per cage under a 12 light/12 dark cycle at a temperature of ~18–23 °C and a 40–60% humidity. Food and water were available ad libitum. To minimize the microbiome variations from cage/rack to cage/rack due to managerial and housekeeping effects, the experimental mice were selected randomly from different cages to house them in one new cage with a maximum number of 5 mice/cage, and then each cage was blindly assigned for different treatments or kept as a control group according to the experimental design. The person who carried out the experiments was not blinded because of the injection of SEB, vehicle, and THC at different times, but most data analyses and experiments were blinded to avoid any bias. The mice were housed in an Association for Assessment and Accreditation of Laboratory Animal Care (AAALAC)-accredited, specific pathogen-free animal facility at the University of South Carolina School of Medicine. All the mouse experiments were performed under protocols approved by the Institutional Animal Care and Use Committee (AUP2363 (old AUP) which is renewed as AUP2500-101498-071720 approved on July 17, 2020). All the studies involving animals are reported in accordance with the Animal Research: Reporting of In Vivo Experiments (ARRIVE) guidelines for reporting experiments involving animals [68,69].

### 4.2. Exposure of Mice with SEB and THC

To induce ARDS, SEB was delivered according to the “dual dose” model described previously [19]. This approach causes a 100% mortality with a low concentration of SEB, and triggers ARDS in C3H/HeJ mice. In brief, SEB dissolved in sterile PBS was administered first by the intranasal (i.n) route at a concentration of 5 μg per mouse in a 25 μL volume. Two hours later, a second dose of SEB was delivered (i.p) at a concentration of 2 μg per mouse in a 100 μL volume. Additionally, THC or vehicle (Ethanol) was administered in 3 doses. The first dose of THC (20 mg/kg, i.p) was given immediately after the first SEB exposure. Then, the second and third doses of THC (10 mg/kg, i.p) were given after an additional 24 h and 48 h, respectively. The SEB-exposed mice that displayed signs of lethargy, hunching, ruffled fur, and respiratory distress were humanely euthanized. The mice were euthanized 72 h post-SEB exposure for tissue collection.

### 4.3. Chemicals and Regents

Staphylococcus enterotoxin B (SEB) was purchased from Toxin Technologies (Sarasota, FL, USA). Delta-9-Tetrahydrocannabinol (THC) was procured from the National Institute on Drug Abuse at the National Institutes of Health (Bethesda, MD, USA). RPMI 1640 culture medium, L-glutamine, penicillin-streptomycin mixture, HEPES buffer, fetal bovine serum, and phosphate buffered saline (PBS) were purchased from Invitrogen Life Technologies (Carlsbad, CA, USA). RNeasy and miRNAeasy Mini kits, miScript primer assays kit, and miScript SYBR Green PCR kits were purchased from QIAGEN (Valencia, CA, USA). The iScript and miScript cDNA synthesis kits were purchased from Bio-Rad (Madison, WI, USA). Epicentre’s PCR premix F and Platinum Taq DNA Polymerase kits were purchased from Invitrogen Life Technologies (Carlsbad, CA, USA).

ELISA kits for IL-2, IL1β, CCL2, and CCL5 (ELISA MAX^TM^ Standard SET Mouse) were purchased from Biolegend. Seahorse XFp Glycolytic Rate Assay Kits and XFP Cell Mito Stress Test Kits were purchased from Agilent Technologies (Santa Clara, CA, USA).

### 4.4. Histopathology of Lung Tissues

Lung tissues from mice were fixed in 4% paraformaldehyde solution, dehydrated in alcohol, and embedded in paraffin. The microtome sections were cut to 5 µm thick, stained with hematoxylin and eosin (H&E), and examined for inflammatory cell infiltrates under Cytation 5 cell imaging multi-mode reader microscopy (Biotek, Winooski, VT, USA).

### 4.5. Analysis of Cytokines×

The cytokine concentrations were measured in broncho-alveolar lavage fluid (BALF). BALF was obtained from euthanized mice by binding the trachea with a suture and excising the lung along with the trachea, as described previously [70]. Then, 1 mL of sterile, ice-cold PBS was injected through the trachea to lavage the lungs. The aspirated BALF was centrifuged to obtain supernatants containing cytokines. ELISAs were performed using ELISA MAX™ standard kits from Biolegend (San Diego, CA, USA), following the manufacturer protocols.

### 4.6. Barrier Function Test

Electric cell-substrate impedance sensing (ECISzθ) system was used to noninvasively real-time monitor the barrier function of lung epithelial cells type II (MLE15) in the presence of SEB-activated splenocytes treated with either THC or Veh for 48 h before being co-cultured with MLE15 cells. Briefly, MLE15 cells were seeded in rate of 12 × 10^5^ cells/well on gold film electrodes arrays, 8W10E+. Once the monolayer of epithelial cells formed, the resistance of this layer was evaluated using a multifrequency test (MFT) to find out what current frequency is suitable, then the resistance was recorded for at least 8 h before coculturing these cells with pre-activated splenocytes with SEB for 48 h and treated with either THC or Veh. The activated splenocytes were washed with RT-PBS twice before being resuspended in 400 µL of complete medium at a concentration of 1 × 10^6^ cells/well. The cells were allowed settle down for 30 min at RT before the re-evaluation of the resistance/capacitance for the next 48 h. A cell-free well was considered as a control, and the barrier resistance values were calculated after being normalized to the resistance of the cells before being wounded with activated splenocytes.

### 4.7. TUNEL Assay

Apoptosis was detected by Apo-Direct terminal deoxynucleotidyl transferase dUTP nick end labeling (TUNEL) Assay Kit (Millipore Sigma, St. Louis, MO, USA). Isolated lung mononuclear cells were washed with PBS, fixed with 4% paraformaldehyde at room temperature for 30 m, and permeabilized with 0.2% TritonX100 for 5 m. The TUNEL solution master mix was added to each tube and incubated in a humidified incubator for 1 h. Next, the cells were washed twice with PBS then analyzed by flow cytometry.

### 4.8. Mitochondrial Membrane Potential

To analyze the mitochondrial membrane potential, lung mononuclear cells were collected and stained with DiOC6(3) dye (ENZO Life Sciences). The cells were incubated in dye for 20 m at 37 °C, then washed twice with pre-warmed PBS and analyzed by flow cytometry.

### 4.9. Effect of THC-Induced MDSCs on T Cell Proliferation In Vitro

To examine the suppressive effect of THC on the T cell proliferation, splenocytes (5 × 10^5^) from C3H naive mice were cultured in the presence of SEB (2 μg/mL) together with different ratios of THC. [3H]thymidine (1 μCi per well) was added to the cell cultures, and, after 18 h, the radioactivity was measured using a liquid-scintillation counter (MicroBeta TriLux; PerkinElmer, Greenville, SC, USA).

### 4.10. Purification of CD3^+^ T Cells

CD3^+^ T cells were purified from splenocytes, as described previously [71]. In brief, the splenocytes were collected from mice and labeled with PE-conjugated anti-CD3 antibody (BioLegend, Clone:17A2). Immunomagnetic selection was achieved by the use of PE Positive Selection Kit (STEMCELL Technologies; Cambridge, MA, USA).

### 4.11. Metabolic Assays

A Seahorse Mito Fuel Flex Test kit and cell mitochondrial stress test kit were used to measure the oxygen consumption rates (OCR) in CD3^+^ T cells according to the manufacturer’s protocol (Agilent, Santa Clara, CA). Purified T cells were seeded in each well of an XFp cell culture mini plate at a density of 2 × 10^5^ cells/well. The XFp cell culture plates were pre-coated with CellTak (Corning, NY, USA) to allow the T cells to adhere to the bottom of wells. Measurements were performed on a Seahorse XFp Analyzer (Agilent, Santa Clara, CA, USA). The Seahorse Wave software was used to interpret the acquired data and calculate the OCR of the purified T cells.

### 4.12. Metabolomic Studies

Measurement of glycolysis, TCA, and Tryptophan metabolites using LC-MS:

Metabolites were extracted from serum using the extraction procedure described previously [72,73,74,75,76,77]. Briefly, 50 µL of the serum sample was used for the metabolic extraction. The extraction step started with the addition of 750 µL ice-cold methanol:water (4:1) containing 20 µL spiked internal standards to each cell pellet or tissue sample. Ice-cold chloroform and water were added in a 3:1 ratio for a final proportion of 1:4:3:1 water:methanol:chloroform:water. The organic (methanol and chloroform) and aqueous layers were mixed, dried, and resuspended with 50:50 methanol: water. The extract samples were deproteinized, followed by resuspension and subjected to 6495 triple quadrupole mass spectrometer (Agilent Technologies, Santa Clara, CA, USA) coupled with a HPLC system (Agilent Technologies, Santa Clara, CA, USA) via single reaction monitoring (SRM).

Separation of Tryptophan metabolites: ESI positive mode was used to measure Tryptophan. For the targeted profiling (SRM), the RP chromatographic method employed a gradient containing water (solvent A) and acetonitrile (ACN, solvent B, with both solvents containing 0.1% Formic acid). The separation of metabolites was performed on a Zorbax Eclipse XDB-C18 column (50 × 4.6 mm i.d.; 1.8 μm, Agilent Technologies, CA) maintained at 37 °C. The binary pump flow rate was 0.2 mL/min, with a gradient spanning 2% B to 95% B over a 25 min time period. Gradient: 0 min—2% B; 6 min—2% of B; 6.5 min—30% B; 7 min—90% B; 12 min—95% B; 13 min—2% B, followed by re-equilibration at the end of the gradient.

### 4.13. Separation of Glycolysis, TCA, and Pentose Pathway Metabolites

Tri carboxylic acid cycle and Glycolysis cycle metabolites were identified by using 5 mM of ammonium acetate in water as buffer PH 9.9 (A) and 100% acetonitrile as buffer (B) using the Luna 3 µM NH2 100 A^0^ Chromatography column (Phenomenex, Torrance, CA, USA). The gradient: 0–20 min—80% B (flow rate 0.2 mL/min); 20–20.10 min—80% to 2% B; 20.10–25 min—2% B (flow rate 0.3 mL/min); 25–30 min—80% B (Flowrate 0.35 mL/min); 30–35 min—80%B (Flow rate 0.4 mL/min); 35–38 min—80% B (flow rate 0.4 mL/min), followed by re-equilibration at the end of the gradient to the initial starting condition of 80% B at a flow rate of 0.2 mL/min. All the identified metabolites were normalized by the spiked internal standard.

### 4.14. Single Cell RNA-Seq and Analysis

The TC20 Automated Cell Counter (BioRad) was utilized to measure the cell count and viability of the isolated lung cells. With a target of 1000 cells, we loaded them onto the Chromium Controller (10 × Genomics). Following the manufacturer’s protocol, the chromium single cell 5′ reagent kits (10 × Genomics) were used to process samples into single-cell RNA-seq (scRNAseq) libraries. The sequencing of those libraries was performed using the NextSeq 550 instrument (Illumina) with a depth of 40–60 k reads per cell. The base call files generated from sequencing the libraries were then processed in the 10 × Genomics Cell Ranger pipeline (version 2.0) to create FASTQ files. The FASTQ files were then aligned to the mm10 mouse genome, and the read count for each gene in each cell was generated. Downstream analysis was completed using Seurat suite version 3.0 [78,79] within R studio. The data were integrated in Seurat using the anchor and integration functions. The integrated data were scaled and a principal component analysis (PCA) was completed for dimensionality reduction. Clusters were made following the PCA analysis by adjusting the granularity resolution to 0.25. We determined the number of principal components (PCs) to utilize the post-JackStraw analysis within Seurat to determine the PCs with the lowest *p*-value. The differential expression was determined for each cluster to determine the cluster biomarkers, and between the disease and treated samples using the default Wilcoxon rank sum test.

### 4.15. Quantitative Real-Time PCR (qRT-PCR) Validation of Selected miRs and Associated Genes

To validate the expression of the selected miR-185 and associated genes (*Bad*, *Bax*, *Cox4*, *Runx3*, *Hrk*, and *Nkiras2*), qRT-PCR was performed in lung-infiltrating MNCs isolated from SEB+Veh or SEB+THC mice. In brief, the total RNA from lung-infltrating MNCs was isolated using the miRNeasy kit from Qiagen and following the manufacturer’s instructions. The miScript primer assays kit and the miScript SYBR Green PCR kit from QIAGEN were used, and the qRT-PCRs were performed following the protocol of the company (QIAGEN, Valencia, CA, USA). For qRT-PCR, we carried out 40 cycles using the following conditions: 15 m at 95 °C (initial activation step), 15 s at 94 °C (denaturing temperature), 30 s at 60 °C (annealing temperature), and 30 s at 70 °C (extension temperature and fluorescence data collection). SNORD96A was used as the housekeeping miR loading control. To determine the expression of genes, *GAPDH* was used as the loading control. Significant differences (*p* < 0.05) in expression were determined by the Student’s *t*-test with an online miRNA database (www.microrna.org). The sequences primers used are described in Table 1.

### 4.16. Correlation of scRNA-seq Data with Human COVID-19 Datasets

To examine the correlation between the SEB+Veh induction of ARDS and COVID-19 disease, we identified dysregulated genes relevant to cytokine and apoptosis from our scRNA-seq data between SEB+Vehicle and SEB+THC and two datasets of RNAseq from the BALF of COVID-19 patients vs. normal controls (https://doi.org/10.1101/2020.04.06.028712. https://doi-org.pallas2.tcl.sc.edu/10.1080/22221751.2020.1747363). Common genes were identified and presented in the format of Venn diagrams.

### 4.17. Data and Analysis

The experimental design and analysis of this study were performed according the recommendations and requirements [80]. A power analysis was carried out (a = 0.05 and 1 − b = 0.80) to estimate the number of animals used in this study, which yielded a minimum sample size of 5 mice per group. Statistical analysis was undertaken only for studies where each group size was at least *n* = 5. All the in vitro studies were carried out in triplicate. All the in vivo studies were performed with at least 5 mice in each group, except for the metabolome analysis (*n* = 4). Statistical analysis was performed using the data generated for individual mice and not using replicates as independent values. The GraphPad Prism 8.0 software was used in the statistical analysis. A Student’s *t*-test was used to compare two groups, whereas multiple comparisons were made using a one-way ANOVA, followed by a post hoc analysis using Tukey’s method. The post hoc test was performed only if we noted an overall statistically significant difference in the group means. All the statistical analyses were carried out using the GraphPad Prism v8 Software (San Diego, CA, USA). A *p* value of <0.05 was considered statistically significant. Each experiment was performed independently at least three times to test the reproducibility of results. The metabolomics data were log2-transformed and normalized with internal standards on a per-sample, per-method basis. Statistical analyses were performed with either an ANOVA or *t*-test in R Studio (R Studio Inc., Boston, MA, USA). Differential metabolites were identified by adjusting the *p*-values for multiple testing at an FDR (Benjamini Hochberg method) threshold of <0.25.

## 5. Conclusions

The current study concludes that the treatment of mice with THC post-SEB challenge protects mice from SEB-mediated toxicity by inhibiting inflammation and ARDS through the modulation of miRs targeting mitochondria-related apoptotic genes. Because SEB is a superantigen that drives cytokine storm, our studies revealed that THC is a potent anti-inflammatory agent that has the potential to be used as a therapeutic modality to treat SEB-induced ARDS. Importantly, the metabolomic and metabolic profiling indicates profound effects on mitochondrial functions that may be responsible for the anti-inflammatory activity of THC. This study also concludes that THC may mediate its effects through downregulating the expression of miR-185-3p and the consequent upregulation of apoptotic genes and pathways; thus, targeting miR-185-3p may constitute another therapeutic modality for the alleviation of ARDS. Using gene expression datasets from the BALF of human COVID-19 patients, we found similarities between the cytokine and apoptotic genes with SEB-induced ARDS. Thus, our data suggests that THC may be useful in treating ARDS and cytokine storm seen in COVID-19 patients.

## Figures and Tables

**Figure 1 ijms-21-06244-f001:**
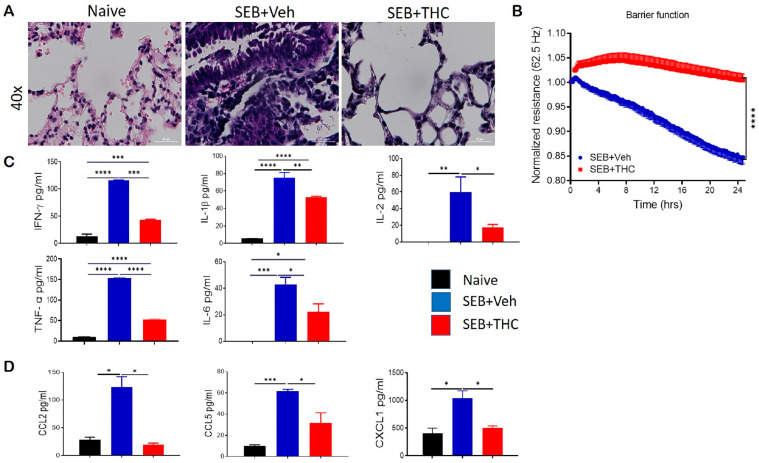
Δ9-Tetrahydrocannabinol(THC) attenuates staphylococcal enterotoxin-B, SEB-induced acute respiratory distress syndrome (ARDS) in mice. For in vivo studies, SEB-mediated ARDS was induced in C3H/HeJ mice, then the mice were treated with either vehicle (Veh) or THC, as described in Methods. For in vitro studies, epithelial cell type II was cultured, and the resistance was measured after adding splenocytes, which were activated with SEB + Veh or SEB+THC. (**A**): Representative hematoxylin and eosin, H&E images of lung tissue sections. (**B**): ECIS measurement of epithelial resistance, presented resistance is normalized to a pre-treatment time point for comparison of THC effect on barrier function. (**C**,**D**): ELISA quantification of broncho-alveolar lavage fluid (BALF) for cytokines (C) and chemokines (D). Vertical bars show data from 5 mice with mean+/-SEM. Statistical significance is depicted as * *p* < 0.05, ** *p* < 0.01, *** *p* < 0.001, and **** *p* < 0.0001 between the groups.

**Figure 2 ijms-21-06244-f002:**
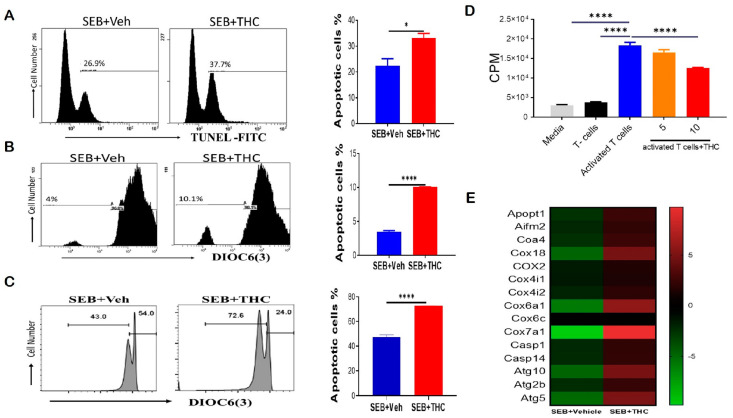
THC attenuates SEB-induced acute respiratory distress syndrome (ARDS) in mice via the induction of apoptosis. For in vivo studies, SEB-mediated ARDS was induced in C3H/HeJ mice, then the mice were treated with either Veh or THC, as described in Methods. For in vitro studies, splenocytes were isolated from naïve mice and activated in culture with SEB+Veh or SEB+THC. (**A**): Terminal deoxynucleotidyl transferase dUTP nick end labeling, TUNEL staining of lung mono-nuclear cells MNCs isolated from SEB+Veh and SEB+THC mice. (**B**): DiOC6(3) staining of CD3+ MNCs isolated from SEB+Veh and SEB+THC mice. (**C**): DiOC6(3) staining of CD3+ splenocytes that were activated in vitro with SEB+Veh or SEB+THC (10 uM). (**D**): ^3^H- Thymidine incorporation assay of splenocytes activated in vitro for 72 h with SEB (1µg/mL) +Veh or SEB+THC (5µm or 10µm). Thymidine incorporation is shown as counts per minute (CPM). (**E**): Heatmap of genes associated with apoptosis in SEB+VEH vs. SEB+THC lung MNCs. Vertical bars in panels A-C show data from 5 mice with mean+/-SEM. Statistical significance is depicted as * *p* < 0.05 and **** *p* < 0.0001 between the groups.

**Figure 3 ijms-21-06244-f003:**
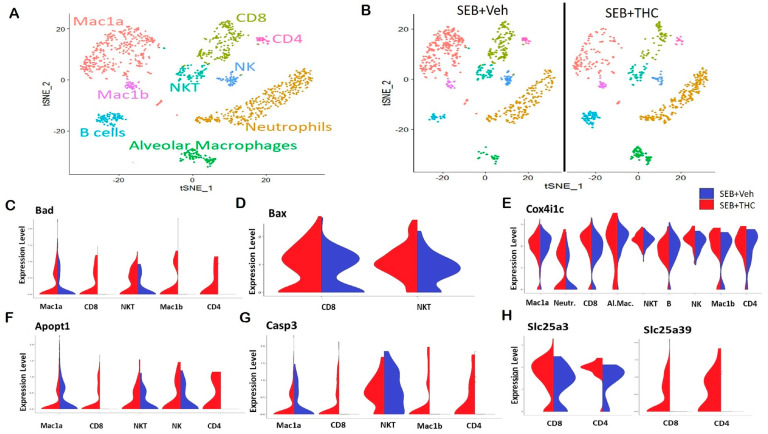
Single-cell RNA sequencing of the lungs reveals that THC upregulates genes associated with apoptosis in SEB-challenged mice. Mice were treated with SEB+Veh or SEB+THC, as described in the Figure 1 legend. (**A**): scRNA-seq t-Distributed Stochastic Neighbor Embedding (tSNE) colored by cell type. (**B**): scRNA-seq tSNE split by sample ID and colored by cell type. (**C**): Violin plots of *Bad* expression amongst the clusters in SEB+Veh vs. SEB+THC. (**D**): Violin plots of *Bax* expression amongst the clusters in SEB+Veh vs. SEB+THC. (**E**): Violin plots of the *Cox4i1c* expression amongst the clusters in SEB+Veh vs. SEB+THC. (**F**): Violin plots of the *Apopt1* expression amongst the clusters in SEB+Veh vs. SEB+THC. (**G**): Violin plots of the *Casp3* expression amongst the clusters in SEB+Veh vs. SEB+THC. (**H**): Violin plots of *Slc25a3* and *Slc25a39* in CD8+ and CD4+ T cells.

**Figure 4 ijms-21-06244-f004:**
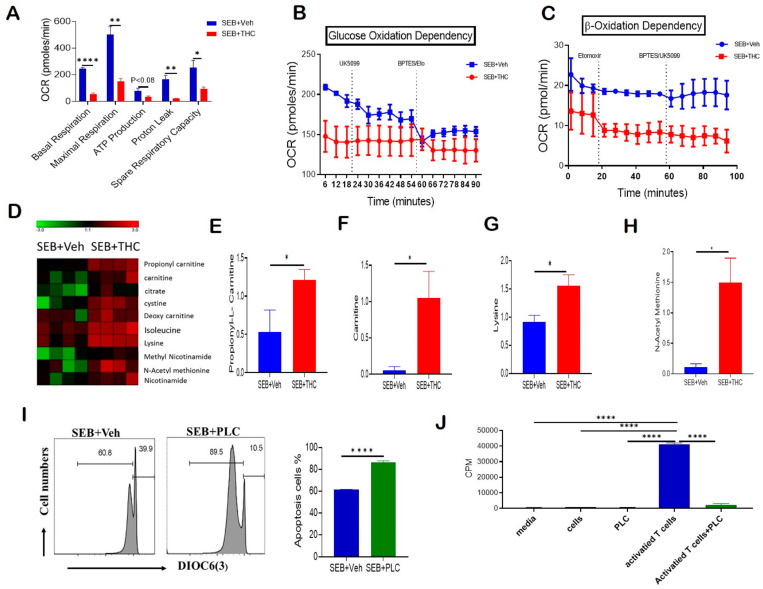
THC treatment decreases T cell activation and alters metabolism to induce apoptosis. Spleen cells were cultured with SEB+THC or SEB+Veh for 72 h, followed by the purification of CD3+ T cells, as described in Methods. (**A**): OCR in cells during the mitochondrial stress test. (**B**): OCR in the glucose oxidation dependency test. (**C**): OCR in the β-oxidation dependency test. (**D**): Heat map showing dysregulated apoptosis-related metabolites from metabolome analysis of serum (*n* = 4). (**E**–**H**): Concentrations of metabolites in serum from SEB+Veh and SEB+THC mice. (**I**): DiOC6(3) staining of splenocytes from C3H/HeJ mice activated with SEB (1 μg/mL) in the presence of either vehicle or PLC 200 μM for 72 h. (**J**): T cell proliferation measured by 3-H thymidine incorporation assay in splenocytes from C3H/HeJ mice activated with SEB (1 µg/mL) in the presence of either vehicle or PLC 200 µm for 72 h. (CPM, counts per minute). Statistical significances are depicted as * *p* < 0.05, ** *p* < 0.01, and **** *p* < 0.0001 between the groups.

**Figure 5 ijms-21-06244-f005:**
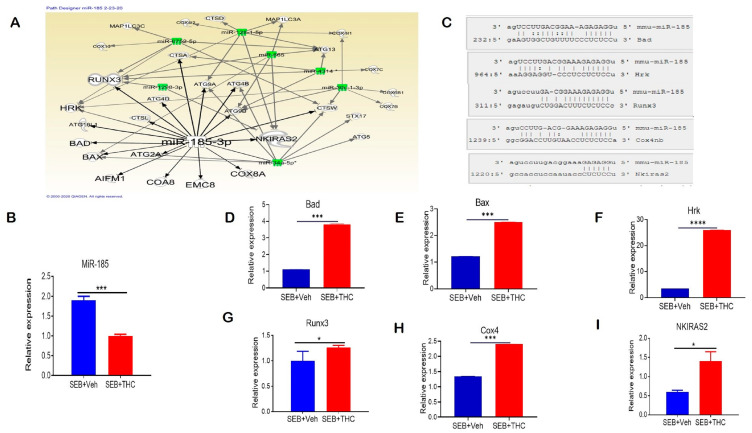
THC treatment in SEB-injected mice results in altered miR expression in lung-infiltrating MNCs. Mice were treated with SEB+Veh or SEB+THC, as described in the Figure 1 legend. MNCs from the lungs were isolated and screened for miR expression. (**A**): IPA pathways and relationships between miRs and genes in lung MNCs. (**B**): qRT-PCR validation of miR-185-3p in lung-infiltrating MNCs. (**C**): Seed sequence alignments between miR-185-3p and *Bad*, *Hrk*, *Runx3*, *Cox4* and *Nkiras2*. (**D**–**I**) qRT-PCR validation of miR-185-3p targeted genes *Bad* (**D**), *Bax* (**E**), *Hrk* (**F**), *Runx3* (**G**), *Cox4* (**H**), and *Nkiras2* (**I**) in lung-infiltrating MNCs. Statistical significance is depicted as * *p* < 0.05, *** *p* < 0.001, **** *p* < 0.0001 between the compared groups.

**Figure 6 ijms-21-06244-f006:**
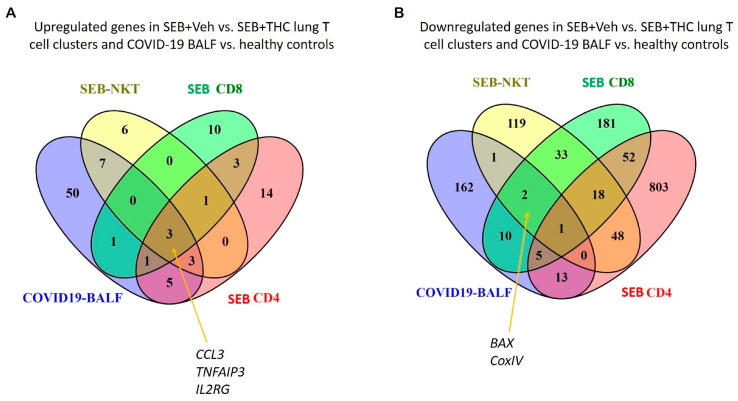
Comparison of SEB-induced cytokines and apoptosis from ARDS mice and COVID19 patients. (**A**): Venn diagram was used in the display of up-regulated genes related to proinflammatory cytokines from SEB-induced ARDS mice and COVID19 BALF patients. (**B**): Venn diagram was used for displaying common down-regulated genes related to the apoptosis pathway from SEB-induced ARDS mice and COVID19 BALF patients.

**Table 1 ijms-21-06244-t001:** Primer sequences.

Gene Name	Forward Primer (5′→3′)	Reverse Primer (5′→3′)
*Runx3*	CAGGTTCAACGACCTTCGATT	GTGGTAGGTAGCCACTTGGG
*Nkiras2*	TCTGTGGGCAAAACTTCAATCC	CGATGGAGCCTACATAGATGTCC
*Bad*	AAGTCCGATCCCGGAATCC	GCTCACTCGGCTCAAACTCT
*Bax*	TGAAGACAGGGGCCTTTTTG	AATTCGCCGGAGACACTCG
*Cox4*	CTCTTCGTGGACTGCATCCC	GGTAATAGCCAGCGATCACATAG
*Hrk*	CAAAGCCTGGGAGGTCTGAG	TCTCACAAAGGCTTCGGTCC

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
