# Peer review of "Δ9-Tetrahydrocannabinol Prevents Mortality from Acute Respiratory Distress Syndrome through the Induction of Apoptosis in Immune Cells, Leading to Cytokine Storm Suppression"

_ijms, 2020, doi:10.3390/ijms21176244_

Round 1

Reviewer 1 Report

This interesting manuscript reports another of the series of experiments showing a beneficial effect of THC in ARDS. Previously, the same group has shown that THC administration prevents mortality induced by SEB  and suggested an immunosuppressive mechanism. Here, the new addition is an insight into the miRNAs and apoptotic pathways. The study can be published after some minor corrections:

  1. please, discuss shortly how the potential of (somehow less controversial) CBD would be in the context of your studies (see for example Khodadadi et al. 2020 https://doi.org/10.1089/can.2020.0043)
  2. The sentence: "Treatment of SEB-mediated ARDS mice with THC led to 100% survival, decreased lung inflammation, and suppression of cytokine storm." is repeated from your F.i.Pharmacology paper. Was the survival estimation performed within this study separately or is it just the same experiment. Please clarify. In the M&M part, I couldn't find any mention about it.
  3. It is difficult (nearly impossible) to find out how many mice per treatment and per a measurement were used (max. 5 mice/cage doesn't mean exactly that it was 5/treatment). It should  be put straightforward in sections 4.1 and 4.2 and in the results as n=X

Author Response

1.please, discuss shortly how the potential of (somehow less controversial) CBD would be in the context of your studies (see for example Khodadadi et al. 2020 https://doi.org/10.1089/can.2020.0043)

We agree with the reviewer and we have discussed the role of CBD (Lines 392-402).  Unfortunately, the paper cited by the reviewer is not yet available on PubMed and only abstract is available online by the journal at this time but not full-length article.  Please note that the CBD tested in this study was in Poly I:C model which triggers cytokine storm but is not lethal unlike our model of SEB-mediated ARDS used in our study, which leads to 100% mortality.  We have found that while CBD also exhibits anti-inflammatory properties it is not as potent as THC. 

2.The sentence: "Treatment of SEB-mediated ARDS mice with THC led to 100% survival, decreased lung inflammation, and suppression of cytokine storm." is repeated from your F.i.Pharmacology paper. Was the survival estimation performed within this study separately or is it just the same experiment. Please clarify. In the M&M part, I couldn't find any mention about it.

We thank the reviewer for pointing this out.  We wish to clarify that these experiments were from the same batch of experiments where we showed that THC can completely prevent SEB-mediated mortality (PMID:  32754917 and 32612530 ).  This has been indicated in the Results section (lines 101-103).    

3.It is difficult (nearly impossible) to find out how many mice per treatment and per a measurement were used (max. 5 mice/cage doesn't mean exactly that it was 5/treatment). It should  be put straightforward in sections 4.1 and 4.2 and in the results as n=X

We thank the reviewer for this question.  Please note that we did provide how many mice were used in M&M under “Data and Analysis” section (lines 564-570) where we stated, “Power analysis was carried out (a=0.05 and 1-b=0.80) to estimate the number of animals used in this study, which yielded a minimum sample size of 5 mice per group for survival studies.  We have also added “All in vivo studies were performed with at least 5 mice in each group, except for metabolome analysis (n=4)” (Line 569-570).  We have also added number of mice in Fig legends. 

Reviewer 2 Report

In the article " Δ9-Tetrahydrocannabinol Prevents Mortality from Acute Respiratory Distress Syndrome through Induction of Apoptosis in Immune Cells Leading to Cytokine Storm Suppression" by Mohammed and the colleague. The authors studied that THC prevents death from acute respiratory distress syndrome by inducing apoptosis of immune cells leading to cytokine storm suppression. With a few modifications, this paper is an interesting finding, and is likely to be published in IJMS.

The manuscript is interesting with valuable findings eventually and has a potential to be published in the IJMS.

  1. What is the exact target for Δ9-Tetrahydrocannabinol (THC)?
  2. Correct the FACS data (Figure 2A-C and Figure 4I) to the same standard. And change it to improve reading and visibility

Author Response

1.What is the exact target for Δ9-Tetrahydrocannabinol (THC)?

THC acts through cannabinoid receptors:  CB1 and CB2. 

2.Correct the FACS data (Figure 2A-C and Figure 4I) to the same standard. And change it to improve reading and visibility.

We used two different flow cytometers which led to different depictions.  We have changed the DiOC6 staining and labeled them all in Y-axis as cell numbers as shown in modified Fig 2C and 4I.